# Development of Cellular Signaling Pathways by Bioceramic Heat Treatment (Sintering) in Osteoblast Cells

**DOI:** 10.3390/biomedicines11030785

**Published:** 2023-03-05

**Authors:** Yoona Jung, Jooseong Kim, Sukyoung Kim, Shin hye Chung, Jinhong Wie

**Affiliations:** 1Department of Physiology, Konkuk University School of Medicine, Chungju 27478, Republic of Korea; 2HudensBio Co., Ltd., 318 Cheomdanyeonsin-ro, Buk-gu, Gwangju 61088, Republic of Korea; 3Department of Biomedical Engineering, Yeungnam University, Daegu 42415, Republic of Korea; 4Dental Biomaterials Science, School of Dentistry and Dental Research Institute, Seoul National University, 101 Daehak-ro, Jongno-gu, Seoul 03080, Republic of Korea

**Keywords:** biocompatible materials, signal transduction, calcium phosphates, osteoblasts

## Abstract

Bioceramics are calcium-phosphate-based materials used in medical and dental implants for replacing or repairing damaged bone tissues; however, the effect of bioceramic sintering on the intracellular signaling pathways remains unknown. In order to address this, we analyzed the impact of sintering on the cell signaling pathways of osteoblast cells using sintered and non-sintered hydroxyapatite (HA) and beta-tricalcium phosphate (β-TCP). X-ray diffraction indicated that only the morphology of HA was affected by sintering; however, the sintered bioceramics were found to have elevated the calcium concentrations in relation to the non-sintered variants. Both bioceramics inhibited the JNK signaling pathway; the sintered HA exhibited half the value of the non-sintered variant, while the sintered β-TCP rarely expressed a p-JNK value. The total Src and Raptor protein concentrations were unaffected by the sintering, while the p-Src concentrations were decreased. The p-EGFR signaling pathway was regulated by the non-sintered bioceramics, while the p-p38 concentrations were reduced by both the sintered β-TCP and HA. All of the bioceramics attenuated the total AKT concentrations, particularly the non-sintered HA, and the AKT phosphorylation concentration, except for the non-sintered β-TCP. Thus, the sintering of bioceramics affects several intracellular signaling pathways. These findings may elucidate the bioceramic function and expand their application scope as novel substrates in clinical applications.

## 1. Introduction

Bone regeneration after fracture is a time-consuming process. Human bone is composed of 60–70% calcium phosphate, which has a chemical composition similar to hydroxyapatite (HA) [1]. Although bone fractures are currently treated using allografts and xenografts, these methods have several disadvantages, such as their high cost, the increased risk of infection and undesirable immune responses [2]. Calcium-phosphate-based bioceramics have therefore attracted considerable attention in recent years as biocompatible replacements for the existing bone graft materials [3]. In particular, HA and beta-tricalcium phosphate (β-TCP), which have chemical compositions similar to those of human bones and teeth, have been shown to accelerate bone regeneration and cell proliferation [4,5,6].

Bioceramics, such as HA and β-TCP, are mainly used as substitutes for hard tissues, such as teeth and bone. However, the mechanical properties of these materials can be further improved through sintering [7]. Sintering is a process that imparts an improved strength and toughness to bioceramics, while also increasing the biocompatibility of the material [8]; this is achieved by densifying or recrystallizing the component atoms of the bioceramic particles through a thermal process [9,10]. Sintering compensates for the naturally low mechanical strength of calcium phosphates and enhances their utility as replacement materials for hard tissues. Additionally, this heating process removes bacteria and other potentially infectious organisms [11,12]. Sintering also enhances the diffusion of round pores and alloying elements in the bulk bioceramic [13]. During high-temperature sintering, the bioceramic microstructure (particle size, shape, porosity ratio, and pore size), chemical composition, and grain boundaries are modified considerably, which directly influences the biological behavior and the mechanical performance of the final material [12,14,15]. However, the traditional sintering method, which involves maintaining high temperatures for long periods of time, has the disadvantage of reducing the porosity of the calcium phosphate microstructure, resulting in increased brittleness and undesirable chemical changes through phase transitions [16].

Sintered HA bioceramics exhibit different biomechanical strengths and physical properties depending on the sintering temperature [17]. Sintering changes can affect the initial bonding behavior of the bioceramic with bone [7,18], as well as induce physiological changes in exposed cells [19]. Additionally, changes in biological behaviors, such as albumin absorption, osteocalcin production, and ALP activity, can be observed depending on the application of sintered and non-sintered HA [20]. Sintering HA at 1200 °C has also been reported to increase the adsorption of bovine serum albumin (BSA) by increasing the bioceramic crystallite size [21]. In contrast, pure β-TCP is generally sintered at temperatures lower than those used for HA; this is because a phase transition occurs from β-TCP to α-TCP at approximately 1150 °C [22,23]. This can enhance the adsorption properties of the material; for example, previous work has shown that the adsorption of MC3T3-E1 on β-TCP can be increased after sintering the material at 1150 °C [24]. Additionally, sintered porous β-TCP has been shown to increase the expression of osteoblast-specific marker genes, ALP, osteocalcin, and interferon-induced transmembrane protein 5, a marker gene involved in osteoblast maturation [25]. As the development of the sintering processing of calcium phosphates progresses, additional research efforts are required to gain better insights into the processes occurring at the tissue–material interface.

Various studies have been conducted to examine the effect of bioceramic surfaces on cell adhesion, spreading, and proliferation; however, the effect of sintering on the intracellular signaling pathways and calcium concentration has not been widely explored thus far. To address this gap in knowledge, we conducted a study investigating the hypothesis that sintering affects the intracellular signal transduction and calcium concentration in osteoblasts. To achieve this, HA and β-TCP were selected as bioceramic substrates, and their effect on these parameters before and after sintering were studied. In the experiments, five signaling pathways (JNK, EGFR, Src, p38, and AKT), which are involved in cell death and proliferation, were analyzed.

## 2. Materials and Methods

### 2.1. Cell Culture

MG63 (osteoblast) cells were purchased from Seoul National University (Seoul, South Korea). The cells were maintained in an atmosphere of 5% CO_2_ at 37 °C in Dulbecco’s Modified Eagle Medium (DMEM, Gibco, Waltham, MA, USA) supplemented with 10% fetal bovine serum and 1× penicillin–streptomycin (Gibco).

### 2.2. Sample Preparation

Non-resorbable HA and resorbable β-TCP were selected as the bioceramic substrates. Both were prepared as sintered and non-sintered powders. First, non-sintered HA (HA-n) was synthesized using octacalcium phosphate (OCP) as the starting material. The synthesis was carried out via OCP hydrolysis, as previously reported [26,27,28]:Ca_8_H_2_(PO_4_)_6_(OH)_2_·5H_2_O + 2Ca^2+^ → Ca_10_(PO_4_)6(OH)_2_ + 3H_2_O + 4H^+^

Next, sintered HA (HA-s) was prepared by heat-treating reagent-grade HA powder (Junsei, Tokyo, Japan) at 1200 °C for 2 h, followed by mechanical crushing to afford a fine powder [29].

Reagent-grade TCP powder (Sigma Aldrich, St. Louis, USA) was used for the non-sintered samples (β-TCP-n). Commercially available β-TCP powder (Himed, Old Bethpage, NY, USA) was used for the sintered samples (β-TCP-s).

### 2.3. X-ray Diffraction (XRD)

To confirm that the prepared powders were suitable for the study, phase analysis was performed using XRD (Aeris, Malvern Panalytical, Malvern, UK) with a Cu-Kα radiation source. Scans were conducted at a rate of 2.0°/min and a step size of 0.022°. The XRD analysis results were compared with the publicly available inorganic crystal structure (ICSD) data.

### 2.4. Field-Emission Scanning Electron Microscopy (FE-SEM)

The surface morphology of the powder samples was examined by FE-SEM (S-4800 Hitachi, Tokyo, Japan) at magnifications of 1000× and 5000×.

### 2.5. X-ray Photoelectron Spectroscopy (XPS)

Elemental analysis of the bioceramic powders was carried out using a Kα XPS spectrometer (K-Alpha, Thermos Scientific, Waltham, MA, USA). Each sample surface was sputter-cleaned with argon ions to remove contaminants before performing the XPS measurements [30]. The elemental compositions in the powders were determined from the C 1s, O 1s, Ca 2p, and P 2p peaks.

### 2.6. Preparation of Bioceramic Plates

Solutions of the bioceramics (10 mg/mL in DMEM) were prepared in 100 pi cell-culture plate dishes (SPL Life Sciences, Pocheon, Korea) and incubated at 37 °C and 5% CO_2_ for 24 h. After incubation, the dishes were used for the cellular studies at 37 °C and 5% CO_2_.

### 2.7. Western Blotting

The signaling proteins JNK/p-JNK, EGFR/p-EGFR, Src/p-Src, Raptor/MTORC1, p38/p-p38, and AKT/p-AKT were examined in the signaling pathway experiments. The proteins were obtained via cell lysis after the cells were incubated in the bioceramic-loaded plates.

First, the cell solutions were centrifuged at 15,000× *g* and 4 °C for 30 min. After centrifugation, the supernatant was removed, and the protein concentration was measured. The proteins were then eluted for 15 min at 70 °C using a 4× lithium dodecyl sulfate sample buffer (diluted in radio-immunoprecipitation assay (RIPA) buffer) supplemented with β-mercaptoethanol. The samples were then separated using 10% sodium dodecyl sulfate-polyacrylamide gel electrophoresis using the mini-Protean Tetra Handcast System (Bio-Rad Laboratories, Hercules, CA, USA). The isolated proteins were then transferred onto polyvinylidene difluoride (PVDF) membranes, which were subsequently blocked with 5% non-fat dry milk in PBS-Tween 20 (PBS-T).

After blocking, the protein-loaded membranes were incubated overnight at 4 °C with the complementary primary antibodies. Subsequently, the membranes were washed five times with PBS-T before further incubation with complementary secondary antibodies. To detect the protein bands, a SuperSignal™ West Femto Maximum Sensitivity Substrate (34095, Thermo Fisher Scientific, Waltham, MA, USA) was used. The primary antibodies used were anti-EGFR (1:1000, 2232S), anti-phospho-EGFR (1:1000, 2236S), anti-SAPK/JNK (1:1000, 9252S), anti-phospho-SAPK/JNK (1:1000, 9251S), anti-Src (1:1000, 2108S), anti-phospho-Src (1:1000, 2101S), anti-p38 (1:1000, 9212S), anti-phospho-p38 (1:1000, 4511S), anti-AKT (1:1000, 9272S), and anti-phospho-AKT (Ser473) (1:1000, 9271S). The HRP conjugated secondary antibodies used for western blot were anti-mouse IgG and anti-rabbit IgG (used at 1:4000 dilution). All of the antibodies were purchased from Cell Signaling Technology (USA.). Anti-Raptor antibodies (1:1000, sc-81537) were purchased from Santa Cruz Biotechnology, Inc. (Dallas, TX, USA).

### 2.8. Inhibitors

SP600125 (10 μM, Selleckchem, Houston, TX, USA), wortmannin (20 μM, Sigma-Aldrich, St. Louis, MO, USA), and SB203580 (30 μM, Sigma-Aldrich, USA) were used in this study.

### 2.9. Intracellular Calcium Measurements

Poly-L-lysine-coated 12 mm coverslips were used for the attachment of the MG63 cells before and after exposure to the sintered and non-sintered bioceramics. After deposition, the cells were loaded with Fura-2 AM (5 μM) and 0.01% pluronic F-127 in a normal Tyrode (NT) solution at 37 °C for 30 min. The NT solution was composed of 140 mM NaCl, 5 mM KCl, 10 mM HEPES, 10 mM glucose, and 1 mM of MgCl_2_ or CaCl_2_ for each condition. The Fura-2-loaded cells were then transferred to a perfusion chamber on an optical microscope (Eclipse Ti, Nikon, Tokyo, Japan).

To confirm the basal calcium concentration in each sample, the MG63 cells were treated with 10 mM EGTA (lowest calcium concentration) and 10 mM Ca^2+^/ionomycin (highest calcium concentration). For the 340 and 380 nm excitation wavelengths, a high-speed wavelength-switching device (Lambda DG-4, Sutter Instrument, Novato, CA, USA) was used. The MetaFluor image analysis software (Molecular device, San Jose, CA, USA) was used to measure the fluorescence intensity ratio (340:380 nm) of several regions across the Fura-2-loaded cell samples.

### 2.10. Statistical Analysis

All of the protein concentration analyses were performed using ImageJ (National Institutes of Health, Bethesda, Maryland, USA). The results were compared using one-way analysis of variance (ANOVA). A *p*-value of <0.05 was defined as a significant difference, and all of the statistical analyses were represented as the mean ± the standard error of the mean (SEM.).

## 3. Results

### 3.1. Phase Characteristics of the Sintered and Non-Sintered HA β-TCP Crystals

The obtained XRD patterns for the sintered and non-sintered bioceramics are shown in Figure 1A. These patterns are consistent with the previously published ICSD data (HA #00-009-0432, β-TCP #00-009-0169). The XRD patterns of the HA-n and HA-s exhibited an HA phase; however, the non-sintered powder had a lower degree of crystallinity, which may have affected its bioactivity in relation to that of the sintered variant. However, the crystallinity in both the β-TCP-n and β-TCP-s powders was similar; thus, these bioceramics are expected to exhibit similar bioactivity and biodegradation rates.

The particle morphologies of the bioceramics were examined by FE-SEM at 1 k and 5 k magnifications (Figure 1B). Here, the HA-n exhibited an indistinct particle shape, whereas the HA-s exhibited a clear spherical morphology. This can be attributed to the two different heating regimes used to prepare these variants, resulting in different degrees of aggregation, which would lead to their varied crystallinities. In contrast, the two β-TCP powders possessed distinct particle shapes owing to their similar crystallinity.

To investigate the Ca/P molar ratio in each bioceramic, energy dispersive spectroscopy (EDS) was conducted by coupling the EDS system with the FE-SEM instrument. The ratios obtained for HA-n and HA-s were 1.29 and 1.55, respectively, thus indicating a disparity in the Ca/P ratio after sintering. Conversely, β-TCP-n and β-TCP-s showed similar ratios of 1.23 and 1.24, respectively. These results were further corroborated by the XPS measurements, as shown in Figure 2A.

### 3.2. Effect of the Bioceramics on the Intracellular Calcium Concentration

Intracellular calcium is important for regulating the cell signaling pathways; thus, we examined the calcium concentration from the cells after exposure to the sintered and non-sintered bioceramics via fluorescence measurements. It was found that the HA-n exhibited a smaller intracellular calcium concentration than that of the HA-s. The β-TCP powders showed similar results (Figure 2B). This indicates that the absence of sintering negatively affects the intracellular calcium concentration in MG63 cells.

### 3.3. Effect of the Bioceramics on the Regulated JNK Pathway

For the signaling pathway studies, the cells were incubated in solutions of the sintered and non-sintered bioceramics and lysed to obtain the intracellular signaling proteins for western blotting analysis. First, the JNK pathway was examined (Figure 3A). The HA-n almost halved the p-JNK signaling pathway in relation to the control, while the HA-s exhibited a further decrease. More considerable pathway reductions were also observed for the β-TCP-n and β-TCP-s. The JNK/p-JNK inhibitor SP600125 (10 μM) was used to verify the drug-induced inhibition. The JNK protein concentration was not affected by the bioceramic treatment, with or without sintering.

### 3.4. Effect of the Bioceramics on the p-Src Pathway

All of the bioceramics were found to inhibit the p-Src pathway considerably. The HA-s and both of the β-TCP variants suppressed the p-Src protein concentration to approximately zero, whereas the exposure to HA-n resulted in comparatively higher p-Src activity (Figure 3B).

### 3.5. Effect of the Bioceramics on the EGFR Signaling Pathway

With sintering, the HA and β-TCP did not exhibit a difference in the p-EGFR protein concentration in relation to the control sample; however, the non-sintered bioceramics attenuated the p-EGFR signaling pathway considerably. The total amount of EGFR protein did not change with or without sintering (Figure 3C). This indicates that the non-sintered bioceramics decrease the EGFR phosphorylation.

### 3.6. Effect of the Bioceramics on the Raptor/MTORC1 Signaling Pathway

With and without sintering, both ceramics exhibited negligible differences in the Raptor protein concentration in relation to the control sample; thus, these bioceramics do not influence the total Raptor protein concentration (Figure 3D).

### 3.7. Effect of the Bioceramics on the p38 Signaling Pathway

The effect of sintering on the p38 signaling pathway was different depending on the type of bioceramic used; for HA, there was no observable difference in the p38 concentration between the sintered and non-sintered variants, whereas β-TCP-n significantly enhanced the p38 phosphorylation, by approximately 50%, in relation to that of β-TCP-s. The total p38 protein concentrations with HA and β-TCP were similar to that of the control group (Figure 4A).

### 3.8. Effect of the Bioceramics on AKT Phosphorylation

The AKT signaling pathway was individually affected by the two types of bioceramics. While all of the bioceramics decreased the total AKT protein concentration in relation to the control sample, p-AKT (S473) exhibited a higher AKT activity with β-TCP-n in relation to that of β-TCP-s. Conversely, both HA-n and HA-s significantly decreased the AKT protein concentration, but did not exhibit a notable difference in the p-AKT protein concentration between the sintered and non-sintered variants. However, HA-n did exhibit a considerable decrease in the total AKT protein concentration, whereas the other bioceramics gave a similar reduction compared to the control sample (Figure 4B).

## 4. Discussion

The sintering process uses high-temperature heat treatment to modify both the physical and chemical properties of bioceramics. Previous studies have mainly focused on examining the macroscopic changes in these parameters such as the porosity, strength, and brittleness [9,10,13,16,17,22,23]. The changes in the osteoblast behavior, such as attachment, differentiation, and proliferation, on these bioceramics has also been examined in relation to the sintering method or temperature [12,14,15,18,19,20,21,24,25]; however, to the best of our knowledge, the ultimate examination of the changes in the calcium concentration or intracellular signaling pathways induced by biometric sintering has not been conducted widely in previous studies. We therefore examined the potential effects of sintering on these characteristics and subsequently established that the sintering effect differs depending on the type of bioceramic and the signal transduction pathway.

Through XRD and FE-SEM measurements, it was established that HA alters its phase composition after sintering, whereas β-TCP is unaffected (Figure 1); HA-s also contains Mg 1s, while HA-n does not. Similarly, the β-TCP-s exhibited increased amounts of Ru 3d after sintering, whereas the concentrations of the other ions did not change (Figure 2A). Combined with the signaling pathway observations, these results demonstrate that the changes in the physicochemical properties of the bioceramics after sintering may affect the intracellular calcium concentration and cell signaling pathways; however, the methodology used in this study is limited in that it is not possible to specify which of the various physicochemical factors may be responsible. One potential explanation is that a change in the bioceramic’s morphology or intrinsic attributes results in a change in the cellular processes. Recently, it has been reported that polymethylmethacrylate (PMMA) disks for CAD/CAM prostheses exhibit desirable mechanical properties and biocompatibility for osteoblast proliferation [31]; if this is combined with a suitable bioceramic, this may result in suitable substrates for clinical use.

Calcium acts as an intracellular messenger that controls numerous cellular processes, including proliferation, mitosis, neurotransmission, gene transcription, and apoptosis. In previous studies, it has been reported that when appropriate mechanical stimulation is given to osteoblasts, the calcium channels in the cell membrane are activated, resulting in an influx of extracellular calcium into the cells. This increases the intracellular calcium concentration and promotes a corresponding increase in bone production [32]. Our results indicate that sintering does not change the intracellular calcium concentration in relation to a suitable control sample; however, the use of non-sintered bioceramics does decrease the concentration (Figure 2B). Despite this observation, it is not clear by what mechanism the non-sintered bioceramics affect the intracellular calcium concentration; one possibility is that when the bioceramics are dissolved in solution, the leaching of the calcium and phosphate from their structures into the buffer may influence the calcium concentration inside the cells. This may occur at different rates depending on whether the bioceramic is sintered or non-sintered. Alternatively, the characteristics of the bioceramics may affect the ion channels of the osteoblasts. Further research is needed to investigate the relationship between the increase in the mechanical stimulation of the osteoblasts owing to sintering, the intracellular calcium concentration, and the resulting activation of the signaling pathways.

Five major intracellular signaling pathways, JNK, Src, EGFR, p38, and AKT, were analyzed in this study. The MAPK signaling cascade pathway, an important signaling pathway known to be involved in the regulation of osteoblast physiology, plays a major role in several processes, including cell proliferation, death, and differentiation [33]. This signaling pathway consists of several components, including JNK and p38 [34]. JNK signaling is broadly activated by oxidative stress and acts as a regulator for several life cycle processes, including cellular meiosis, mitosis, differentiation, and energy metabolism [35]. In this study, it was confirmed that HA-n slightly decreases the p-JNK signaling activity in relation to other materials; additionally, these pathways can be further inhibited by SP200125 (a JNK inhibitor) (Figure 3A). Moreover, both β-TCP-n and β-TCP-s decrease the p-JNK protein concentration considerably, while leaving the total JNK protein concentration intact.

Src activity has been reported to maintain bone homeostasis and regulate cell invasion and growth [36,37]. Its class, tyrosine kinases, also affects the morphology, survival, and cell proliferation of osteoblasts [38]. Additionally, Src suppresses the activity of RUNX2, a crucial regulator of osteoblast development that decreases the osteoblast’s ability to regenerate bone [39]. In a previous study, Src was shown to disrupt osteoblast proliferation, which explains why cell proliferation is enhanced by the bioceramics used in this study [40]. Regardless of sintering, the bioceramics were found to decrease the p-Src protein concentration, particularly when β-TCP was used (Figure 3B).

EGFR is characteristic of multiple carcinomas and significantly affects the cell signaling pathway [41]. In osteoblasts, EGFR activates the proliferation and inhibits the differentiation of osteoblasts [42]. Our results indicate that the sintered bioceramics do not affect the p-EGFR protein concentration (Figure 3C) in the MG63 cells. However, the non-sintered bioceramics exhibited a considerable effect without changing the total protein concentration. Despite this, further studies are required to examine the effect of the bioceramic physicochemical properties on this pathway.

We also examined the effect of the total Raptor protein concentration on cell proliferation; however, this was not impacted by the bioceramics, regardless of sintering (Figure 3D).

p38 is a member of the MAPK family and is associated with accelerating cell differentiation and proliferation [43]. The treatment with the bioceramics increased the p-p38 concentration, particularly for β-TCP-n; in comparison, β-TCP-s resulted in a lower increase. Compared to β-TCP, HA resulted in lower p-p38 concentrations (Figure 4A). This indicates that each type of bioceramic affects this signaling pathway differently.

β-TCP-n does not influence AKT phosphorylation, whereas β-TCP-s slightly decreases the p-AKT concentration. HA also decreases the p-AKT concentration, regardless of sintering (Figure 4B). All AKT signaling pathways are further inhibited by wortmannin (a AKT inhibitor). Notably, all of the bioceramics decreased the total AKT concentration, particularly HA-n, by a factor of five. These results indicate that each type of bioceramic individually influences this cellular signaling pathway, and that sintering has a considerable effect. However, the exact mechanisms behind this effect are unclear and require further study.

Through this study, it was demonstrated that the sintering process alters various physicochemical factors in the bioceramics, which can affect the intracellular calcium concentration and signal transduction pathways in osteoblasts. Recently, it has been reported that exacerbated intracellular signaling results in cleft lip, palate, and alveolus (CLPA) defects in children and adolescents [44]. Bioceramics are biomaterials that play an important role in improving the future quality of these treatments and promoting wound healing. As shown in the results of this study, bioceramics can act as factors that affect various intracellular signal transduction; this is because the effect on the intracellular signaling pathways is different depending on the type and material processing. It is necessary to have an in-depth understanding of the signaling pathways and intracellular calcium concentrations that can activate the signaling pathways. However, there is a limit to proving the relationship between these factors. In future studies, it is necessary to specify the physical factors that are altered by sintering and to ascertain which of these factors individually affect the calcium concentration in osteoblasts and the activation of their signaling pathways.

## 5. Conclusions

In this study, several bioceramics were shown to differently regulate the intracellular signaling pathways and intracellular calcium concentration based on whether the substrates were sintered. Bioceramics may play a significant role in improving the future quality of these treatments and promoting bone regeneration. We investigated the effects of bioceramics on cells and how to improve the utilization of calcium phosphate-based bioceramics. Our findings suggest that the employed bioceramics exhibit individual effects on the regulated intracellular signaling pathways. The results of this study enhance the understanding of the intracellular response to bioceramics used in combination with various biomaterials to promote bone regeneration and can be used as basic data for the development of biomaterials containing bioceramics in the future. Bioceramics may play a significant role in improving the future quality of these treatments and promote wound healing. It is expected to help to improve the quality of clinical research. In future studies, it is necessary to examine the specific physicochemical changes induced by bioceramic sintering for an in-depth understanding of the effect on the intracellular calcium concentration and signal transduction pathway of osteoblast cells.

## Figures and Tables

**Figure 1 biomedicines-11-00785-f001:**
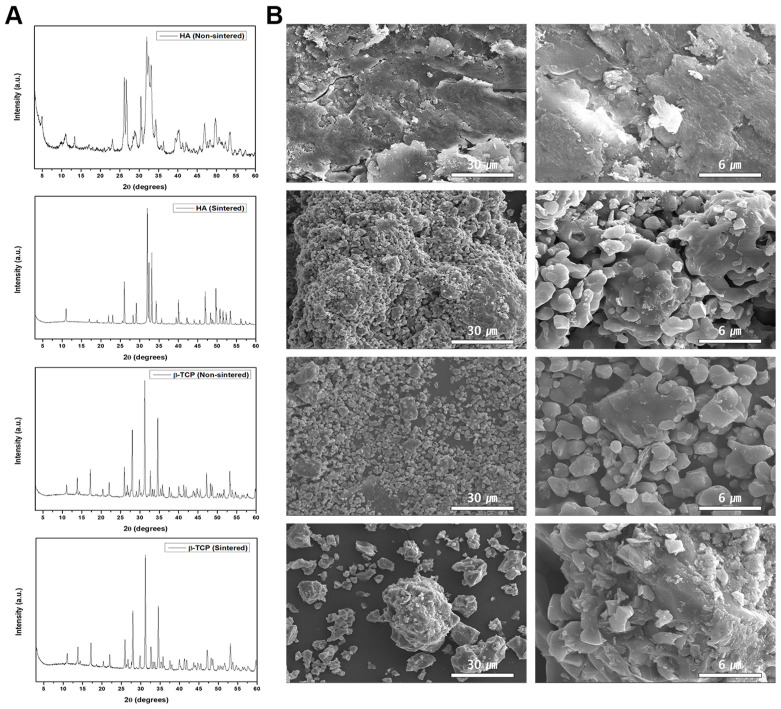
(**A**) X-ray diffraction (XRD) patterns and (**B**) field-emission scanning electron microscopy (SEM) images of sintered and non-sintered hydroxy apatite (HA) and beta-tricalcium phosphate (β-TCP). Scale bar = left(30 μm), right(6 μm).

**Figure 2 biomedicines-11-00785-f002:**
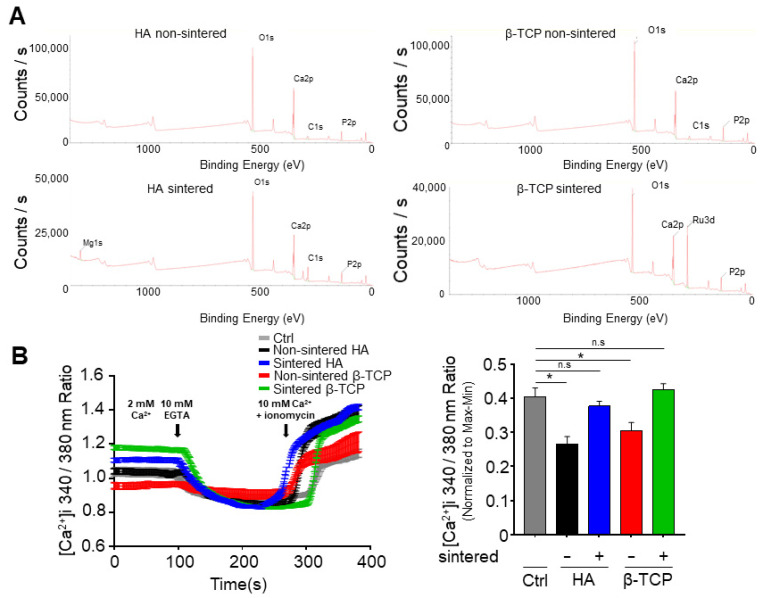
(**A**) X-ray photoelectron spectroscopy (XPS) spectra of sintered and non-sintered HA and β-TCP; (**B**) Intracellular calcium measurements from cells incubated with the sintered and non-sintered bioceramics. The bar graph shows values presented as a mean ± the standard error of the mean (SEM.). The values were also normalized to those of the control sample. The results were compared using the one-way analysis of variance (ANOVA) method. * *p* < 0.05; n.s = not significant.

**Figure 3 biomedicines-11-00785-f003:**
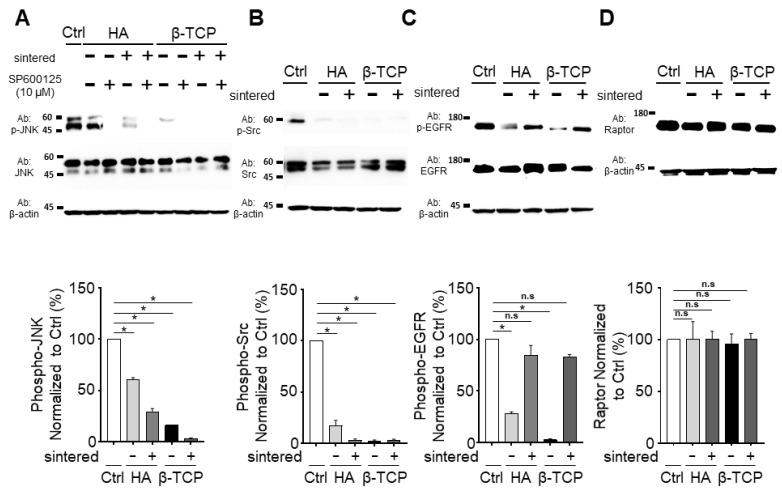
Western blotting measurements examining several intracellular signaling pathways of MG63 cells after exposure to the sintered and non-sintered bioceramics: (**A**) JNK/p-JNK; (**B**) Src/p-Src; (**C**) EGFR/p-EGFR; (**D**) Raptor. The bar graphs show values presented as the mean ± the SEM., which have been normalized to the control samples in each case. The results were compared using the one-way ANOVA method. * *p* < 0.05; n.s = not significant.

**Figure 4 biomedicines-11-00785-f004:**
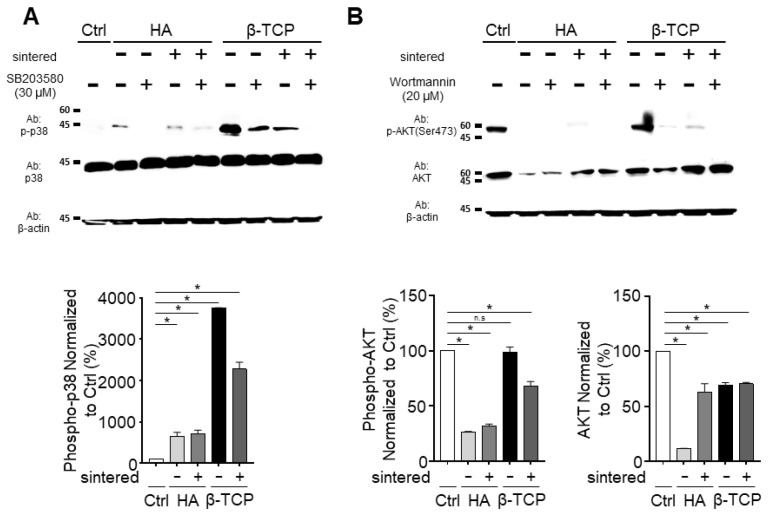
Western blotting measurements examining the p38 and AKT pathways of MG63 cells after exposure to the sintered and non-sintered bioceramics: (**A**) p38/p-p38; (**B**) AKT/p-AKT. The bar graphs show values presented as the mean ± the SEM., which have been normalized to the control samples in each case. The results were compared using the one-way ANOVA method. * *p* < 0.05; n.s = not significant.

## Data Availability

The data presented in this study are provided in this article.

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
