# Peer review of "Development of Cellular Signaling Pathways by Bioceramic Heat Treatment (Sintering) in Osteoblast Cells"

_biomedicines, 2023, doi:10.3390/biomedicines11030785_

Round 1

Reviewer 1 Report

It is a paper on the efficacy and mechanism of bioceramics and shows the role of bone regeneration. I think it will be a good reference paper for the development of materials used in dental work.

Reviewer 2 Report

Generally interesting study with however many exposition gaps especially in the introduction and discussion part.

In any case, I consider the work acceptable only after profound modifications listed below

- An initial sentence is missing in the abstract section on the general issue that led to the study

-Line 10 define, listing them, the tests used in the study

- A sentence is missing at the end of the abstract section that brings possible clinical implications of the study

-Check that all keywords are pubmed MESH terms

-Line 35 to define, for the greater interest of the reader, the possible fields of application in the medical and dental fields of repair with bioceramics

-line 49 lacks bibliographic reference

-The purpose of the study must be expressly indicated at the end of the introduction section, also inserting the null hypotheses of the same which will be refuted at the end of the results obtained

-The introduction section is too meager in my opinion and does not address the problems with due in-depth analysis.

-paragraph 2.6 too meager to define incubation and times

-Line 228 lacks bibliographic reference

-Some considerations on the biocompatibility aspects of these substitutes should be added to the discussions. In this regard, I ask you to add this scientific work to the reference section which could be of help to the reader:

Pagano S, Lombardo G, Caponi S, et al. Bio-mechanical characterization of a CAD/CAM PMMA resin for digital removable prostheses. Dent Mater. 2021;37(3):e118-e130. doi:10.1016/j.dental.2020.11.003

-Another aspect to consider are the possible applications of these substitutes in patients with congenital defects. In this regard, I ask you to insert in the reference section the following scientific work that could be of help:

Pasini M, Cagidiaco I, Fambrini E, Miceli M, Carli E. Life Quality of Children Affected by Cleft Lip Palate and Alveolus (CLPA). Children (Basel). 2022;9(5):757. Published 2022 May 21. doi:10.3390/children9050757

-A section on the limitations of the study is missing

Reviewer 3 Report

Introduction. This investigation is an study that presents information for researchers in the field of bone regeneration with bioceramic materials. Calcium hydroxyapatite (HA) and beta-tricalcium phosphate (β-TCP) are widely used to accelerate bone regeneration. When these bioceramics are applied to the human body, they are mainly used as a substitute for bone tissues. The effect of sintering on intracellular signaling pathways is still being determined. In this study, we hypothesize that sintering affects both the material composition and the signaling pathways.

Last paragrah must incorporate some references.

The aim of the study must be clearly determined.

Materials and methods.

This in vitro study was designed to analyze the characteristics and properties of bioceramic materials  about bioactivity and biodegradation rates.

This section showes a good structure of different subsections (i.e. cell culture, simple preparation, X-ray diffraction, SEM, XPS), but the author must explain if this protocol is original or is based in before published experiments. In fact, the authors must incorporate some references about every subsection for to improve the scientific evidence of methodology.

Results

This section showes a good structure of different subsections (characteristics of the bioceramic materials, effect on intracellular calcium level, signalling pathways) and include several figures related with results.  

The results must be report only the main experimental findings. However, several paragraphs showes aspects of methodology and discussion:

Methodology:

Calcium is crucial for the intracellular signaling pathway. To evaluate the effects of 152 sintered and non-sintered bioceramics on intracellular calcium levels, we examined 153 whether bioceramics influenced intracellular calcium levels in MG63 cells. To confirm ba- 154 sal calcium level, we treated the MG63 cell with 10 mM EGTA (lowest calcium level) and 155 10 mM Ca2+ + ionomycin (highest calcium level).

Discussion:

Non-sintered HA inhibited the p-Src protein level, whereas sintered HA dramatically sup- 179 pressed it. β-TCP also modulated the Src signaling pathway. It inhibited the p-Src protein 180 level with or without sintering. This indicated that bioceramics negatively regulated the 181 p-Src pathway (Figure 3B).

Considering the p-p38 protein 203 levels without bioceramics, all bioceramics effect (Figure 4A). However, the total protein 204 p38 levels did not change. These results indicate that heat treatment and bioceramics pos- 205 itively regulated the cell signaling pathways.

To evaluate the effect of bioceramics on the AKT signaling pathway, we incubated 213 MG63 cells with and without bioceramics (10 mg/ml) and treated with AKT inhibitor 214 wortmannin (20 μM) after 3 days. The cell signaling pathway was clearly affected. This 215 indicated that AKT activity individually affects the two types of bioceramics with and 216 without sintering.

Discussion.

This section must include the analysis of results according the scientific evidence of similar studies. This section only includes two long paragraph. Also, the discussion includes an small number of related references. These references must be updated. Only two references are published at last 5 years.

Conclusions. This section is very long and includes some aspects of results and discussion

References. This section is very short and must be increased with new references. Many references are older. The references include only 6 papers (30%) of last five years

Conclusively, the study is not ready for publication.

Round 2

Reviewer 2 Report

All comments were added 

Reviewer 3 Report

The section Conclusions is very long and includes some aspects of results and discussion
